# Comparative Metabolomics Profiling of Polyphenols, Nutrients and Antioxidant Activities of Two Red Onion (*Allium cepa* L.) Cultivars

**DOI:** 10.3390/plants9091077

**Published:** 2020-08-21

**Authors:** Rita Metrani, Jashbir Singh, Pratibha Acharya, Guddadarangavvanahally K. Jayaprakasha, Bhimanagouda S. Patil

**Affiliations:** 1Vegetable & Fruit Improvement Center, Department of Horticultural Sciences, Texas A&M University, 1500 Research Parkway, Suite A120, College Station, TX 77845-2119, USA; ritametrani@tamu.edu (R.M.); singh2014@tamu.edu (J.S.); pratibha@tamu.edu (P.A.); 2National Center of Excellence for Melon at the Vegetable and Fruit Improvement Center of Texas A&M University, College Station, TX 77845, USA

**Keywords:** *Allium cepa* L., honeysuckle, sweet Italian, flavonoids, anthocyanins

## Abstract

Onion is among the most widely cultivated and consumed economic crops. Onions are an excellent dietary source of polyphenols and nutrients. However, onions phytonutrient compositions vary with cultivars and growing locations. Therefore, the present study involved the evaluation of polyphenol, nutritional composition (proteins, nitrogen, and minerals), sugars, pyruvate, antioxidant, and α-amylase inhibition activities of red onion cultivars, sweet Italian, and honeysuckle grown in California and Texas, respectively. The total flavonoid for honeysuckle and sweet Italian was 449 and 345 μg/g FW, respectively. The total anthocyanin for honeysuckle onion was 103 μg/g FW, while for sweet Italian onion was 86 μg/g FW. Cyanidin-3-(6”-malonoylglucoside) and cyanidin-3-(6”-malonoyl-laminaribioside) were the major components in both the cultivars. The pungency of red onions in honeysuckle ranged between 4.9 and 7.9 μmoL/mL, whereas in sweet Italian onion ranged from 8.3 to 10 μmoL/mL. The principal component analysis was applied to determine the most important variables that separate the cultivars of red onion. Overall results indicated that total flavonoids, total phenolic content, total anthocyanins, protein, and calories for honeysuckle onions were higher than the sweet Italian onions. These results could provide information about high quality and adding value to functional food due to the phytochemicals and nutritional composition of red onions.

## 1. Introduction

Onions are among the most widely grown, consumed, and important economic crops [1]. They are consumed alone or in prepared foods for their unique pungent flavor and sugar content [2]. According to the U.S Statista, the consumption of fresh onions accounted for 20.4 pounds per capita in 2019 [3]. Onions are excellent sources of polyphenols, contribute significantly to the human diet, and have been used in the prevention of infections, cancer, cardiovascular disease, and hyperlipidemia [4,5].

Polyphenols are receiving increasing interest from consumers and food manufacturers due to antioxidant and anti-inflammatory benefits that help prevent illnesses and diseases. Polyphenols are natural substances in plants and include compounds such as phenolic acids and flavonoids [6]. Flavonoids are further categorized into six different subclasses: flavonols, flavones, isoflavones, flavanones, anthocyanins, and flavanols, forming the largest group with antioxidant effects [7]. Health benefits of phenolics attract the researcher and breeders to identify and develop new cultivars with enhanced functional properties [8,9]. Selective breeding of red onion with the phenolic-rich content and high antioxidant activity can provide various health benefits [10].

Onions are a rich source of flavonoids, mainly quercetin mono- and diglucosides, kaempferol, isorhamnetin, and myricetin [11]. Flavonoids possess numerous health beneficial properties such as anticarcinogenic, anticholesterol, antidepressant, antifungal, antidiabetic, antioxidant, etc. [12]. Nishimura et al. reported that the daily intake of onion rich in quercetin might improve liver function and have anti-obesity effects [13]. Due to the structural functionalities, quercetin has maximum radical scavenging activity and acts as a potent antioxidant, representing the main antioxidative flavonol in the human diet [14]. Antioxidants have the capacity to prevent and slow the oxidation of molecules. These oxidation reactions produce free radicals, thus causing damage to cells [15]. Flavonoids in onion have strong antioxidant activity due to the presence of hydroxyl groups, which may help to prevent coronary heart disease and tumors [16]. Flavonoid contents vary in the different layers (outer paper layer, first flesh layer, second flesh layer and inner flesh layer) of onion, and results showed that significantly higher flavonoid content was observed in the first layer as compared to others layers [17]. A short-day red onion (TX 90977) was reported to contain 101.2 μg/g of total quercetin [18]. The southern Italian red onion had two major flavonoids such as quercetin-4′-glucoside (208 to 230 μg/g) and quercetin-3,4′-diglucoside (254 to 274 μg/g FW) [19].

The red variety of onions is mainly due to the presence of pigment anthocyanins. Anthocyanins are hydrophilic pigments which belong to the subclass of flavonoids. They are glycosides of anthocyanidin and responsible for the red or blue coloration in fruits, flowers, and other plant parts [20,21]. Anthocyanins are potent free radical scavengers and have demonstrated protection against oxidative DNA cleavage, enzyme inhibition, oxidative degradation of lipids, and membrane strengthening [14]. They have anticancer, anti-inflammatory, and antidiabetic activities. They also improve cardiovascular health and prevention against Alzheimer’s and Parkinson’s diseases [22,23,24]. In onion germplasm, cyanidin derivatives are the major while others such as pelargonidin and delphinidin derivatives are the minor anthocyanins [25].

Onions are also rich in minerals (potassium and iron), vitamins, and dietary fiber [15]. The effect of onion phenolic phytochemicals on hyperglycemia has been investigated. Previous studies reported that polyphenols such as quercetin, naringenin, catechins, epicatechins, chlorogenic acids, ferulic acids, caffeic acid, and tannic acids had α-amylase inhibiting effects [26,27]. The inhibition of α-amylase enzyme prevents the breakdown of polysaccharides to monosaccharides resulting in the lowering of the blood sugar level [28]. Therefore, phenolic enriched plants such as onions may possess many therapeutic benefits.

The aim of this research was to quantify the phytonutrients present in two varieties—honeysuckle and sweet Italian onions—which are the most widely consumed red onions in the United States. To our knowledge, there is no study reporting the content of phytonutrient (flavonoids, anthocyanins, total phenolics), antioxidant activity (ABTS, DPPH), α-amylase inhibition activity, and pungency levels in these cultivars.

## 2. Results and Discussion

Honeysuckle red onions from J&D Produce, Inc., Texas and Artisan sweet Italian red onions from Tanimura & Antle, California, were used for the entire study (Figure 1).

### 2.1. Pungency, Total Soluble Solids, and Sugars

Maceration of onion tissue causes generation of pyruvic acid, ammonia, and sulfur related compounds due to the reaction of the enzyme alliinase with (S)-alk(en)yl cysteine sulfoxide. The sulfur compounds as an unstable and pyruvic acid as a stable compound have been used as an indicator of onion pungency [29,30]. The pungency of onion is presented in Figure 2A. The mean pyruvic acid content ranged between 6.3 (honeysuckle) and 9.1 (sweet Italian) μmoL/mL. There was a significant variation between the red onions of two cultivars. According to Liguori et al., the level of pungency is as follows: low, 0–3 μmoL/mL; moderate, 3–7 μmoL/mL; high, >7 μmoL/mL [31]. The Brix of two cultivars was 10.9 (honeysuckle) and 11 (sweet Italian) and did not show any significant difference. The red onions are generally regarded as high in pungency and Brix.

The glucose, fructose, and sucrose were detected in all the onion samples (Figure 2B). The glucose (19.8 mg/mL) and fructose (25.2 mg/mL) were high in sweet Italian, whereas the sucrose (11.29 mg/mL) was high in the honeysuckle variety. The total sugar ranged between 42.13 (honeysuckle) and 53.68 (sweet Italian) mg/mL.

### 2.2. Evaluation of Protein and Minerals

Onion serves as a good supplement of magnesium, calcium, iron, sodium, phosphorous, boron, and potassium. Although onions constitute a small portion of the whole diet, they play an important role in metabolic processes and normal functioning of the human body. The results obtained for the mineral and protein content analyzed in all the samples and their differentiation, according to the cultivars are shown in Table 1. Phosphorous helps to keep bones strong and repair damaged tissues [32]. The highest level of phosphorous was seen in sweet Italian cultivar (2677.64 μg/g) and the lowest in honeysuckle (2525.63 μg/g). Potassium plays an important role in maintaining water balance, regulation of heartbeat, and neurotransmission [33]. The highest level of potassium (13,550.1 μg/g) was observed in sweet Italian and lowest (12,720.37 μg/g) in honeysuckle.

Magnesium helps in the carbohydrate metabolism, a cofactor in more than 300 enzymatic reactions, stabilizes protein, and calcium absorption for bones [34]. The average consumption of magnesium for men is 420 mg and women is 320 mg. The values for magnesium in sweet Italian and honeysuckle were 1100.62 and 980.43 μg/g, respectively. Calcium, sodium, zinc, copper, sulfur, and boron were high in honeysuckle with the values 3183.52, 1001.34, 14.43, 7.38, 3421.58, and 18.34 μg/g, respectively. Sodium regulates the body’s water content and helps in the absorption of certain nutrients, while iron is essential for the formation of hemoglobin in red blood cells [35]. Zinc acts as a cofactor for enzymes and helps in reproductive development [36], whereas copper is important for infant growth and producing red and white blood cells. Manganese is a constituent of antioxidant enzyme which helps to prevent free radical-mediated damage to cells. Boron helps the body to metabolize vitamins and minerals [37], whereas sulfur helps to build and repair DNA and protect cells against damage [38]. The protein percentage in honeysuckle (9.21) was higher than the sweet Italian (8.2). Minerals play an essential role from building strong bones to transmitting nerve impulses. There were differences between the concentrations of all the mineral elements studied in the two red onion cultivars. It can, therefore, be assumed that the environment, agronomic practices, and genetic information of the seeds determine the changes in the protein and mineral element composition.

### 2.3. Antioxidant Activity, Total Phenolic Content, and α-Amylase Inhibitory Activity

The DPPH assay is applicable to hydrophobic compounds, whereas ABTS is applicable to both hydrophilic and hydrophobic compounds which provide a flexibility to be used for samples extracted at different pH levels. DPPH and ABTS free radicals react with chemicals compounds by single electron and hydrogen atom transfer mechanisms, respectively [39,40]. The antioxidant activity was measured using DPPH and ABTS assays (Figure 3A,B). There was no significant difference in DPPH activity in sweet Italian onion and honeysuckle with the value of 262.93 and 259.75, respectively. ABTS radical scavenging activity was high in sweet Italian onions (193.82 μg/g) compared to honeysuckle onions (173.27 μg/g). Sagar et al. [41], Wu et al. [42], and Lu et al. [43] reported 66.79–94.17 μg GA/g, 103.91 μg AA/g, and 1301.508 μg Trolox/g FW DPPH radical scavenging activity in red onions.

Antioxidants can inhibit oxidative reactions and help in the functional performance of enzymes for self-defense mechanisms within the cell [43]. The consumption of onions has been reported to lower the risk of neurodegenerative disease, cancers, cataract formation, and ulcer development due to antioxidant effects [44].

The α-amylase activity of red onions was determined and compared with a positive control, acarbose. The inhibition of α-amylase showed a significant difference between both the varieties: sweet Italian was high (46.1%) compared to honeysuckle (Figure 3C). The total phenolic contents of the two varieties are shown in Figure 3D. The content value in honeysuckle was 1.57 mg GAE/g, while sweet Italian was 1.40 mg GAE/g fresh weight. Prakash et al. [45] reported the total phenolic content in the outer, middle, and inner layers of red onion to be 74.1, 15.9, and 5.6 mg GAE/g dry weight basis. In another study, Yang et al. reported that the total phenolic content in Northern red onions was 0.815 mg GAE/g fresh weight [46].

### 2.4. Quantification of Red Onion Flavonoids

The HPLC-PDA chromatographic separation and elution pattern of different flavonoids are shown in Figure 4A. The identification of onion flavonoids was performed by using LC-HR-ESI-QTOF-MS in positive ionization mode. A total of eight flavonoids was identified, and their tandem mass spectra with +bbCID spectra and the structures of the identified flavonoids are presented in Figure 5 and Figure 6, respectively. The accurate mass and mass error of the identified onion flavonoids are presented in Table 2. A peak eluted at retention time (t_R_) 10.6 min showed an accurate mass spectrum at *m*/*z* 789.2013 [M+H]^+^ (mass error 8.91 ppm). The precursor ion lost a glucose molecule to give a product ion at *m*/*z* 465.1010 [M+H-162-162]^+^.

It further lost one glucose molecule to give a prominent base peak at *m*/*z* 303.0481 [M+H-465-162]^+^ (Y_0_)^+^ with a mass error of 6.03 ppm. Thus, the present peak was identified as quercetin-3,7,4′-triglucoside based on the mass spectra and published literature [11]. Another peak eluted at (t_R_) 16.5 min representing the molecular ion peak at *m*/*z* 627.1507 [M+H]^+^ (mass error 7.74 ppm), and aglycone product ion at *m*/*z* 303.0476 (mass error 7.68 ppm) was recognized as quercetin-7,4′-diglucoside. Similarly, a prominent peak (t_R_ 17.5 min) was identified as quercetin-3,4′-diglucoside with an precursor ion at *m*/*z* 627.1527 [M+H]^+^ (mass error 4.44 ppm) and product ion at *m*/*z* 465.0994 [M+H-162-162]^+^ and a aglycone ion at *m*/*z* 303.0491 (Y_0_)^+^ with a mass error 2.73 ppm. A minor peak at (t_R_) 18.8 min displayed an accurate mass at *m*/*z* 641.1657 [M+H]^+^ (mass error 8.5 ppm). The precursor ion lost two molecules of glucose to give a prominent product ion at *m*/*z* 317.0631 [M+H-162-162]^+,^ which was attributed to aglycone isorhamnetin moiety. Thus, the peak at 27.1 min displayed a similar precursor ion at *m*/*z* 465.1012 [M+H]^+^ (mass error 3.16 ppm) and at *m*/*z* 465.1038 [M+H]^+^ (mass error −2.29 ppm), respectively. Both precursor ions lost one molecule.

Glucose gave a product ion at *m*/*z* 303.0482 [M+H-162]^+^ and *m*/*z* 303.0510 [M+H-162]^+^, respectively. Thus, the present peaks were identified as quercetin-3-glucoside and quercetin-4′-glucoside, respectively. A peak at (t_R_) 27.1 min was identified as isohamnetin-4′-glucoside with a precursor ion at *m*/*z* 479.1201 [M+H]^+^ (mass error -3.70 ppm) and a product ion at *m*/*z* 317.0662 [M+H-162]^+^. Similarly, the mass spectrum of a minor peak eluted at (t_R_) 29.3 min showed a precursor ion at *m*/*z* 303.0496 [M+H]^+^ (mass error 0.92 ppm) which corresponded to the presence of aglycone quercetin in the onion methanol extract.

Table 3 shows the comparative levels of flavonoids, respectively, in the onion cultivars. There was no significant difference between cultivars in these individual flavonoids measured except isorhamnetin-4′-glucoside and quercetin-4′-glucoside. Table 3 shows that honeysuckle cultivar had the highest amount of isorhamnetin-4′-glucoside and quercetin-4′-glucoside with values of 25.5 and 184.6 μg/g of fresh onions, respectively, as compared to sweet Italian onions with values 13.52 and 115.3 μg/g of fresh onions, respectively.

In our study, we found that quercetin-3,4′-diglucoside and quercetin-4′-glucoside were the most abundant flavonoids, and these results are in agreement with the previous studies [11,19,47]. Overall, the total flavonoids were higher in honeysuckle cultivar (440.93 μg/g) than sweet Italian (338.39 μg/g). Serban et al. [48] reported the quercetin effect on blood pressure in 7 trials comprising 9 treatment arms (587 patients). The authors reported that supplementation with quercetin >500 mg/day causes significant reductions in both systolic and diastolic blood pressure. Quercetin glucosides possess antioxidant, anti-inflammatory, cardioprotective, and anti-allergic properties. Quercetin also reduces the risk of clot formation near the damaged endothelium [49].

### 2.5. Identification and Quantitation of Anthocyanins

In the present study, seven anthocyanins were identified using LC-HR-ESI-QTOF-MS in positive ionization mode. The protonated accurate mass and mass error of the identified anthocyanins are presented in Table 2. The tandem mass spectra with +bbCID spectra and the structures were identified.

Anthocyanins are presented in Figure 7 and Figure 8, respectively. A peak eluted at retention time (t_R_) 13.5 min exhibited a UV spectrum at 520 nm, which corresponded to characteristic flavylium cation. A MS spectrum showed an accurate mass at *m*/*z* 449.1022 [M]^+^ (mass error 12.4 ppm). The precursor ion lost a glucose molecule to give a product ion at *m*/*z* 287.0516 [M-162]^+^ (mass error 11.8 ppm), which corresponded to the aglycone moiety. Thus, based on the MS and +bbCID mass spectra, UV-Vis-spectra, and published literature [50,51,52], the present peak was identified as cyanidin-3-glucoside. Similarly, another peak eluted at t_R_ 15.8 min represented the molecular ion peak at *m*/*z* 611.1549 [M]^+^ (mass error 9.3 ppm) and product ion at *m*/*z* 287.0524 [M-162-162]^+^ (mass error 9.0 ppm) was identified as cyanidin-3-laminaribioside.

HR-QTOF-MS spectra of the precursor ion (t_R_ 24.9 min) showed an accurate mass at **m*/*z** 627.1496 [M]^+^ (mass error 9.4 ppm). The precursor ion lost two molecules of glucose to give a prominent product ion peak at *m*/*z* 303.0474 [M-162-162]^+^ (mass error 8.2 ppm), which was attributed to aglycone delphinidin moiety. Thus, the present peak was identified as delphinidin-3,5-diglucoside. Two peaks eluted at 31.8 and 32.6 min displayed a molecular ion at *m*/*z* 535.1024 [M]^+^ (mass error 10.8 ppm) and at *m*/*z* 697.1532 [M]^+^ (mass error 11.7 ppm), respectively. Both peaks were identified as cyanidin derivatives. A peak (t_R_ 31.8 min) showed a prominent product ion at *m*/*z* 287.0531 [M-162-86]^+^ (mass error 10.8 ppm) due to the loss of glucose and a malonyl moiety. Similarly, another peak (t_R_ 32.6 min) lost two molecules of glucose and a malonyl moiety and displayed an intense base peak at **m*/*z** 287.0527 [M-162-162-86]^+^ (mass error 11.1 ppm). Thus, the present peaks were identified as cyanidin-3-(6”-malonylglucoside) and cyanidin-3-(6”-malonyl-laminaribioside), respectively.

A minor peak (t_R_ 33.4 min) was identified as peonidin-3-malonylglucoside with a precursor ion at *m*/*z* 549.1168 [M]^+^ (mass error 12.7 ppm) and a product ion at *m*/*z* 301.0665 [M-162-86]^+^ (mass error 13.6 ppm). The MS and +bbCID spectra of the peak eluted at t_R_ 33.9 min displayed a precursor ion at *m*/*z* 577.1124 [M]^+^ with a mass error of 10.9 ppm. The product ion was at *m*/*z* 287.0518 [M-162-86-42]^+^.

The mass error 11.1 ppm was yielded with a loss of glucose, malonly, and acetyl molecules. On the basis of the accurate mass, fragmentation pattern, and literature reports, the present peak was identified as cyanidin-3-(malonyl)-(acetyl)-glucoside.

Figure 4B and Table 3 show the HPLC chromatograph and comparative levels of anthocyanins, respectively, in the onion cultivars. Cyanidin-3-(6”-malonyl-laminaribioside), cyanidin-3-laminaribioside, and peonidin-3-malonoylglucoside, which were 30.1 ± 2.1 μg/g, 5.3 ± 0.4 μg/g, and 1.0 ± 0.08 μg/g, respectively, were significantly higher in honeysuckle onion. The total anthocyanin content was higher in honeysuckle (103 μg/g FW) than sweet Italian (86 μg/g FW). Other authors, such as Pérez-Gregorio et al. [53], Ferreres et al. [54], and Gennaro et al. [55], reported the total anthocyanins in red onions to be 28.68, 233, and 90 μg/g, respectively. Anthocyanins exert different biological effects, and consumption of food rich in anthocyanins may be useful for antidiabetic, anti-obesity, neuroprotective agents, cardiovascular protection, and inhibiting cancer growth [56].

### 2.6. Principal Component Analysis and Pearson’s Correlation Coefficients

The principal component analysis (PCA) model was applied to determine the essential variables that separate the cultivars of sweet red onion. The first (PC1) and the second (PC2) components explained 40% and 23.3% of the total variance, respectively (Figure 9). PCA grouped samples into two cultivars that reflected marked differences in the phytochemical depending on the cultivar. The corresponding biplot emphasized that the separation between the two cultivars was characterized by flavonoids, anthocyanins, sucrose, total phenolic content, DPPH, and minerals (Ca, Na, B, Cu, Zn) which were high in honeysuckle red onions. However, the amount of pyruvic acid contributed to the distinction between the two cultivars.

To seek the correlation between phytochemical contents and antioxidant activities, Pearson’s correlation coefficient was observed (Appendix A). DPPH showed a high positive correlation with quercetin, quercetin-7,4′-diglucoside, quercetin-3,7,4′-triglucoside, quercetin-3-glucoside, quercetin-4-glucoside, and isorhamnetin-3,4-diglucoside (r > 0.70, 0.75, 0.77, 0.55, 0.54, and 0.64 respectively). The protective effect of flavonoids is due to their capacity to transfer hydrogen free radicals, act as scavengers of free radicals, activate the antioxidant enzyme, and inhibit oxidases [57]. The antioxidant activity of quercetin-3-glucoside and quercetin-4′-glucoside on iron ion-driven lipid peroxidation of the gastrointestinal mucosa in rats was studied. In this study, the rat gastrointestinal mucosa homogenates were incubated with Fe(NO3)_3_, ascorbic acid, quercetin, and its glucoside where quercetin-4′-glucoside suppressed the mucosal lipid peroxidation, which turned out to be favorable antioxidant sources [58]. Anthocyanin such as delphinidin-3,5-diglucoside, cyanidin-3,6-malonoylglucoside, and cyanidin-3-glucoside was strongly correlated with DPPH with Pearson coefficient of 0.75. Cyanidin-3-glucoside possesses strong antioxidant activity due to two hydroxyls on the B-ring, and recent studies support its bioactivity as DNA-RSC, gastroprotective, anti-inflammatory, antithrombotic, chemopreventive, and as an epigenetic factor [59]. Li et al. reported the protective effect of cyanidin-3-glucoside by lowering the cardiovascular complications on type 2 diabetes rat model [60]. Similarly, a glucoside derivative of cyanidin showed significant inhibition of α-amylase and sucrase enzymes which may help in the prevention of diabetic complications [61]. Our result shows that the correlation between α-amylase and cyanidin-3-glucoside was found to be 0.45. In addition, the quercetin and anthocyanin derivatives were slightly correlated to the total phenolics (<0.5). On the other hand, ABTS was weakly associated with the quercetin derivatives and anthocyanins group.

## 3. Material and Methods

### 3.1. Plant Materials and Chemicals

Honeysuckle red onions from J & D Produce, Inc., Texas and Artisan sweet Italian red onions from Tanimura & Antle, California were used for the entire study (Figure 1). Sodium pyruvate, 2, 4-dinitrophenylhydrazine (DNPH), glucose, sucrose, fructose, quercetin 3-glucoside, Quercetin-3,4′-diglucoside, quercetin, delphinidin 3-glucoside, L-ascorbic acid, gallic acid, dextrose, sodium carbonate, 2,2-diphenyl-1-picryhydrazyl, 2,2′-azino-bis-(3-ethylbenzothiazoline-6-sulphonic acid), Folin-Ciocalteu reagent, HPLC grade methanol, acetonitrile, formic acid, and metaphosphoric acid were purchased from Sigma Aldrich (St. Louis, MO, USA). Sodium hydroxide (NaOH) was obtained from Fisher Scientific (Pittsburg, PA, USA).

### 3.2. Pyruvic Acid Content

The neck, basal plate, and skin of the onion bulbs were removed and blended to extract onion juice. The pyruvic acid content in the onion juice was recorded by an automated dinitrophenyl hydrazine method previously developed by our group [62,63]. Soluble solid content (SSC) was measured in the juice using a refractometer and expressed as °Brix.

### 3.3. Sugar Analysis

The sucrose, glucose, and fructose were analyzed using HPLC and an RI detector. A total of 20 μL of the sample was injected into the HPLC system which was connected to a binary pump (Perkin Elmer LC-200, Norwalk, CT, USA), an autosampler (Perkin Elmer LC-200), a refractive index-150 detector (SystemSpectra), and a carbohydrate column Rezex™ RCM-Monosaccharide Ca^2+^ (8%) of 300 × 7.8 mm (Phenomenex, Torrance, CA, USA) equipped with a guard cartridge. The column temperature was maintained at 80 °C using a column heater (Jones chromatography, Lakewood, CO, USA). The water was used as a solvent at a flow rate of 0.6 mL/min that was degassed using degasser (Gastorr TG-14). Sugar concentrations were calculated using standard curves for sucrose, glucose, and fructose ranging from 1.25 to 20 mg/mL.

### 3.4. Protein and Mineral Analysis

Approximately 45 g pulp was stored at −80 °C for 24 h and lyophilized (Labconco Freeze-Dryer, Kansas City, MO, USA) for two days to obtain the dried powder. The dried samples were weighed, and the protein and mineral content was measured at the Texas AgriLife Extension Soil, Water, and Forage Testing Laboratory. Inductive coupled plasma-atomic emission spectroscopy (ICP-AES) (Spectro Genesis, Deutschland, Germany) was used to estimate the mineral content of onion.

### 3.5. Analysis of Total Phenolic Content, Antioxidant Capacity, and α-Amylase Assay

Total phenolics, DPPH, and ABTS radical scavenging activity of both onion cultivars were determined using our previously published methods [64,65]. Fresh onion samples (15 g) were extracted twice with 15 mL methanol. The extracts were pooled together and further used for quantification of total phenolics and free radical scavenging activities. All standards and samples were pipetted in triplicate into 96 well plates separately. The absorbance was measured using Synergy™ HT Multi-Mode Microplate Reader (BioTek, Instruments, Winooski, VT, USA) at 760, 515, and 734 nm for total phenolics, DPPH, and ABTS assays, respectively. The results of total phenolics were expressed as mg gallic acid equivalent/gram of fresh weight of samples, and radical scavenging activities were expressed as mg ascorbic acid equivalent/gram of dry weight sample.

The α-amylase inhibitory activity was measured using a published protocol with minor modifications [66]. Dextrose was used to prepare a standard curve at different concentrations. Acarbose was used as a positive control and methanol as a negative control to calculate the percentage inhibition.

### 3.6. Analysis of Flavonoids

Onion pulp (10 mg) was weighed and extracted with 20 mL methanol, vortexed for 15 s, sonicated for 30 min, and then centrifuged at 4480× *g* for 10 min. After decanting the filtrate, the residue was re-extracted with 20 mL methanol followed by the above procedure. The two extracts were combined, filtered, and transferred to an amber HPLC vial. An Agilent 1200 Series HPLC (Agilent, Foster City, CA, USA) system consisting of a degasser, quaternary pump, autosampler, column oven, and a diode array detector. The separation was carried out using an RP C18 Gemini series column (250 × 4.6 mm; 5 μm) (Phenomenex, Torrence, CA, USA). A 5 μL sample was injected into the column with a flow rate of 0.8 mL/min, the oven temperature was set at 35 °C, and the peaks were monitored at 210 and 280 nm with a run time of 30 min. The elution was carried out using gradient mode elution with acetonitrile (A) and 0.03 M phosphoric acid (B). Initially, elution was carried out at 98–75% B for (0–12 min), followed by 75–45% B (12–22 min), 45–5% B (22–26 min), 5–98% B (26–29 min) and returned to 98% B at 30 min. The column was equilibrated for 2 min before the next injection. Quantification of flavonoids was carried out using standards quercetin-3-glucoside, quercetin-3,4′-diglucoside, and quercetin. Flavonoid results were expressed as μg/g of fresh weight of the sample.

### 3.7. LC-HR-ESI-QTOF-MS Identification of Flavonoids

Flavonoids were identified in onion extract using Agilent 1290 liquid chromatography (Santa Clara, CA, USA) coupled to a maXis Impact high-resolution mass spectrometer (Bruker Daltonics, Billerica, MA, USA) according to our published paper [67]. Briefly, a chromatographic resolution was achieved by column Zorbax Eclipse plus (1.8 μm; 50 × 2.1 mm) with a gradient solvent system, (A) 0.1% formic acid in water, and (B) 0.1% formic acid in acetonitrile with a flow rate of 0.2 mL/min at 30 °C. Elution of flavonoids was performed by programming as follows: 2–25% B (12 min), 25–55% B (11 min), 55–98% B (4 min), 98–2% B (2 min), and 2 min isocratic at 2% B. The injection volume was 1 μL and postrun equilibrium was 1 min. Mass spectral analysis was performed by electrospray ionization (ESI) in positive ionization mode. Mass spectra (MS) and broadband collision-induced dissociation (bbCID) acquisition were acquired in the *m*/*z* 50–2000 scan range. Nitrogen was used as a nebulizer and drying gas. A nebulizer gas pressure and drying temperature was 3.2 bar and 250 °C, respectively, and the capillary voltage was kept at 3500 V. The external calibration of the mass spectrometer was performed by sodium formate solution clusters using the high-precision calibration mode. Software Data Analysis 4.3 was used for the determination of accurate mass for the molecular ions and precursors ions.

### 3.8. Analysis of Anthocyanins

Anthocyanin extraction was performed according to published paper [68]. Freshly cut red onion (10 g) was extracted with acidic methanol (1% formic acid). The sample was homogenized (1 min), sonicated for 30 min, and centrifuged at 7826× *g* under dark conditions. The supernatant was transferred to the new tubes, and residue was re-extracted twice with acidic methanol. All the supernatants were pooled, volume was measured, and the sample was kept at −80 °C for further analysis.

Anthocyanins were analyzed by using Waters 1525 HPLC (Milford, MA, USA) equipped with a 717 plus autosampler and a 2996 photodiode array detector (PDA). Anthocyanin separations were achieved by a Zorbax Eclipse plus C18 ODS column (250 × 4.6 mm) with particle size of 5 μm (Agilent, Santa Clara, CA). The mobile phase consisted of (A) 0.03 M phosphoric acid in water and (B) acetonitrile/water (50:50, *v*/*v*) with a flow rate of 0.6 mL/min. The gradient elution was as follows: 25% B isocratic (3 min), 25–40% B (7 min), 40% B isocratic (6 min), 40–80% B (2 min), 80–25% B (2 min), and 25% B isocratic (4 min). The chromatogram was acquired at 520 nm, and data were processed by Empower-2 software. Anthocyanins were expressed as μg/g of fresh weight of the sample.

### 3.9. LC-HR-ESI-QTOF-MS Identification of Onion Anthocyanins

Anthocyanins were analyzed by 1290 Agilent LC system (Agilent, Santa Clara, CA, USA). A C18 column, rapid resolution high definition (RRHD) Zorbax Eclipse plus (1.8 μm; 50 × 2.1 mm) (Agilent, Santa Clara, CA, USA), was used for the chromatographic separation. The gradient mobile phase consisted of A) 0.1% formic acid in water and (B) 0.1% formic acid in acetonitrile. Mass spectral analysis was performed according to the published papers [69]. Briefly, a maXis impact mass spectrometer (Bruker Daltonics, Billerica, MA) was used for obtaining the mass spectra in ESI in positive ionization mode. MS and bbCID data were obtained at *m*/*z* 50–2000 scan range. Nitrogen was used as a nebulizer (gas pressure of 2.1 bar) and drying gas (8.0 L/min). The capillary ion voltage was 4500 V, and the drying gas temperature was 250 °C. External mass spectrometer calibration was performed by sodium formate solution. The determination of accurate mass data for the molecular ions was performed using the software Data Analysis 4.3.

### 3.10. Statistical Data

Each onion variety had 15 bulbs of an average weight of 170 g and was divided into five groups. The analysis was carried out on five grouped samples belonging to each onion variety. Each experiment was performed in triplicate, and all data are presented as the mean ± SD. Analysis of variance (ANOVA) was carried out using JMP pro 14 software. Excel software was used to prepare graphs, and multiple mean comparisons (*p* value  <  0.05) were carried out by a Student’s *t*-test. Principal component analysis (PCA) was performed by SIMCA-P+ software (version 16, Umetrics AB, Umea, Sweden) using the Pareto scaling method.

## 4. Conclusions

This study provides information describing the nutritional values in honeysuckle and sweet Italian red onions. The pungency of honeysuckle was lower than the artisan onions. The total flavonoid for honeysuckle and artisan onions was 449 and 345 μg/g FW, respectively. The total anthocyanin for honeysuckle onion was 103 μg/g FW, while for artisan onion was 86 μg/g FW. Honeysuckle onions showed relatively high levels of calcium, sodium, zinc, and copper along with nitrogen and protein content. The results indicated that the total flavonoids, total phenolic content, total anthocyanins, protein, and calories for honeysuckle onions were higher than the artisan onions. The PCA model was applied to all data to determine the most important variables that separate the cultivars of red onion. Pearson’s correlation coefficients among quercetin derivatives and DPPH were higher than 0.7; while, quercetin derivatives and total phenolics were 0.5. Overall, these findings suggest that these varieties are good for consumption, with most of the nutritional values required for the human diet.

## Figures and Tables

**Figure 1 plants-09-01077-f001:**
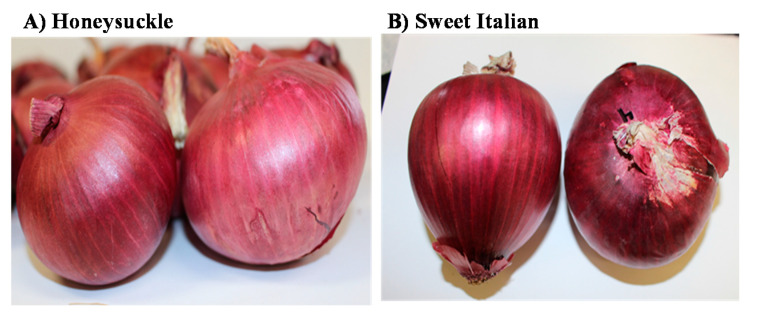
Onion varieties (**A**) honeysuckle and (**B**) sweet Italian.

**Figure 2 plants-09-01077-f002:**
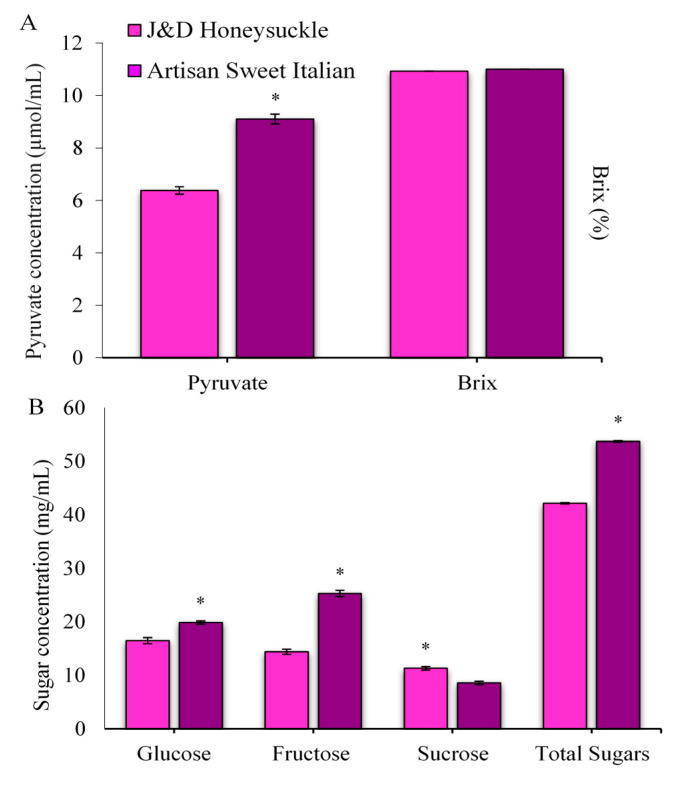
(**A**) Pyruvate concentration and Brix of red onion juice; (**B**) sugar concentration of red onion juice. * Above the bars of each cultivar indicate significant differences as determined by a Student’s *t*-test at *p* < 0.05.

**Figure 3 plants-09-01077-f003:**
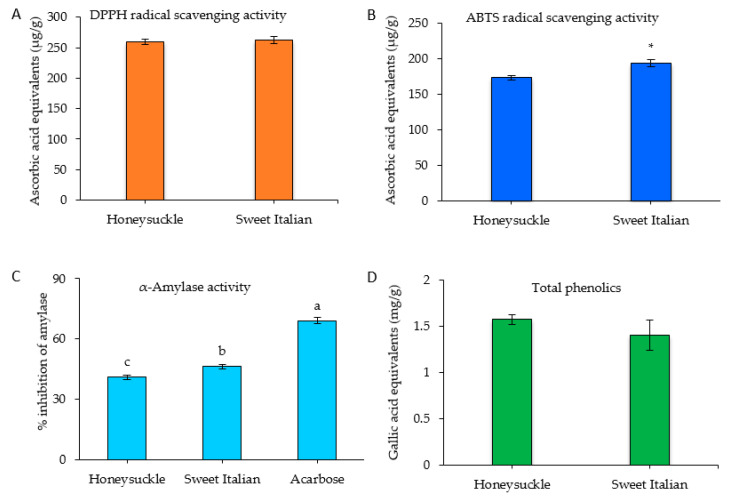
(**A**) ABTS, (**B**) DPPH, (**C**) α-amylase activity, and (**D**) total phenolic content of red onion juice. * Different letters (a, b, c) above the bars of each cultivar indicate statistically significant differences determined by a Student’s t-test at *p* < 0.05.

**Figure 4 plants-09-01077-f004:**
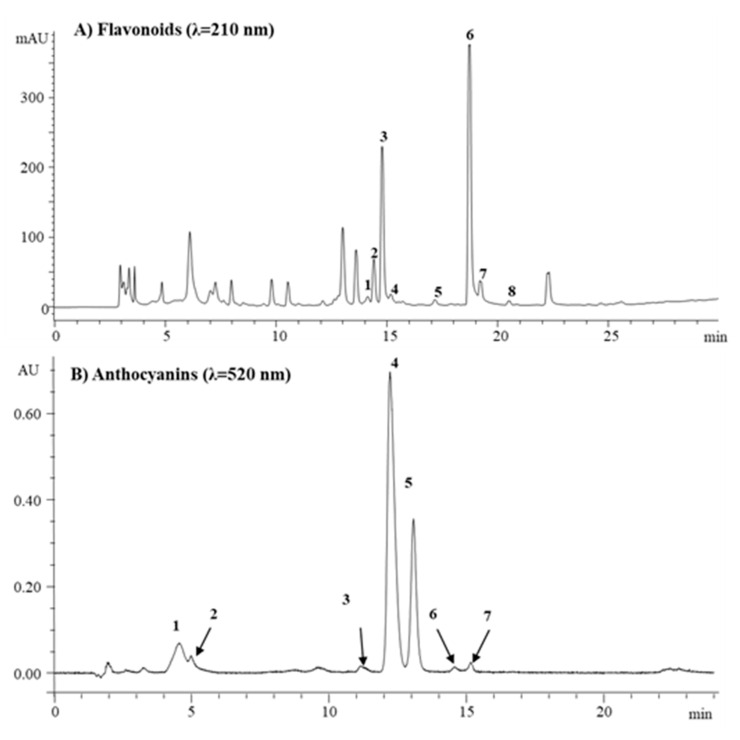
(**A**) Chromatogram of onion flavanoids. Components 1–8 were identified as follows: 1: Quercetin-3,7,4′-triglucoside; 2: Quercetin-7,4′-diglucoside; 3: Quercetin-3,4′-diglucoside; 4: Isorhamnetin-3,4′-diglucoside; 5: Quercetin-3-glucoside; 6: Quercetin-4′-glucoside; 7: Isorhamnetin-4′-glucoside; 8: Quercetin. 4 (**B**) Chromatogram of onion anthocyanins. Components 1–7 were identified as follows: 1: Cyanidin-3-glucoside; 2: Cyanidin-3-laminaribioside; 3: Delphinidin-3,5-diglucoside; 4: Cyanidin-3-(6”-malonoylglucoside); 5: Cyanidin-3-(6”-malonoyl-laminaribioside); 6: Peonidin-3-malonoylglucoside; 7: Cyanidin-3(malonoyl)-(acetyl)-glucoside.

**Figure 5 plants-09-01077-f005:**
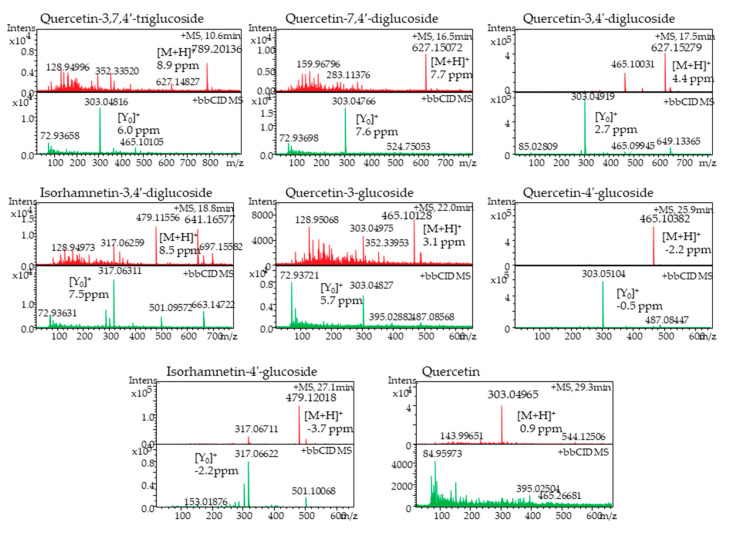
Red onion flavonoids mass spectra identified in positive-ionization mode by LC-HR-ESI-QTOF-MS.

**Figure 6 plants-09-01077-f006:**
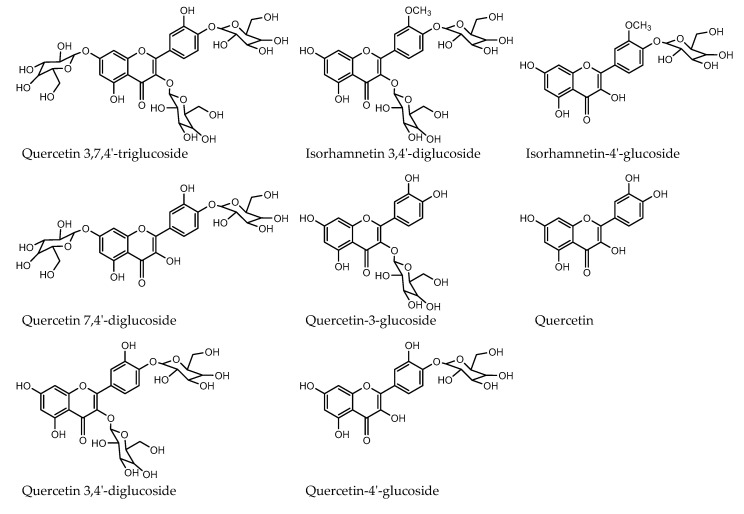
Structure of identified flavonoids from red onion.

**Figure 7 plants-09-01077-f007:**
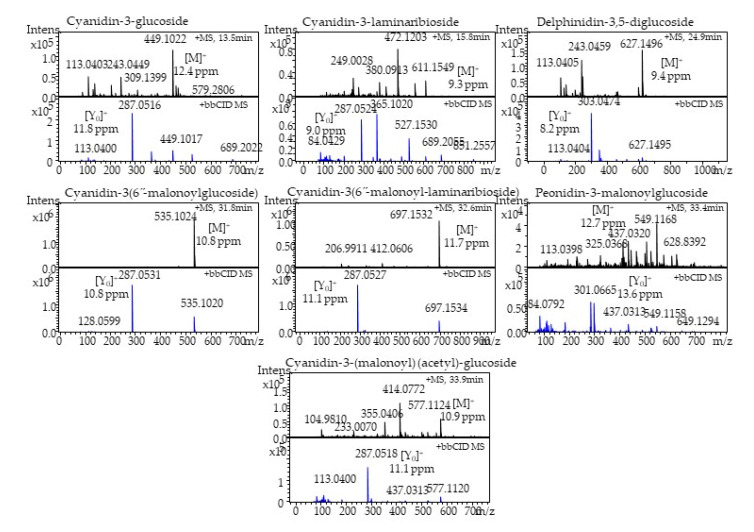
Seven major anthocyanins identified from red onion by LC-HR-ESI-QTOF-MS.

**Figure 8 plants-09-01077-f008:**
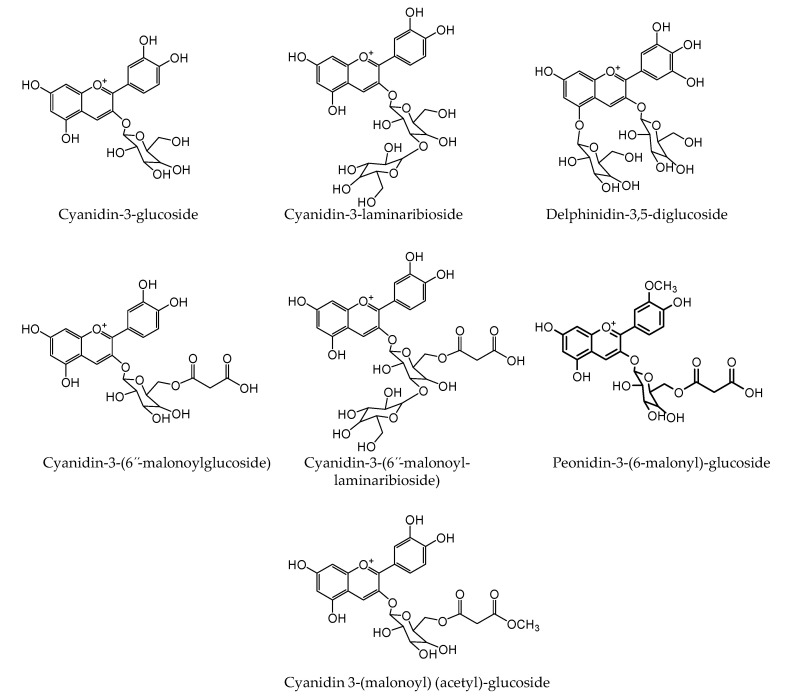
Structure of identified anthocyanins from red onions.

**Figure 9 plants-09-01077-f009:**
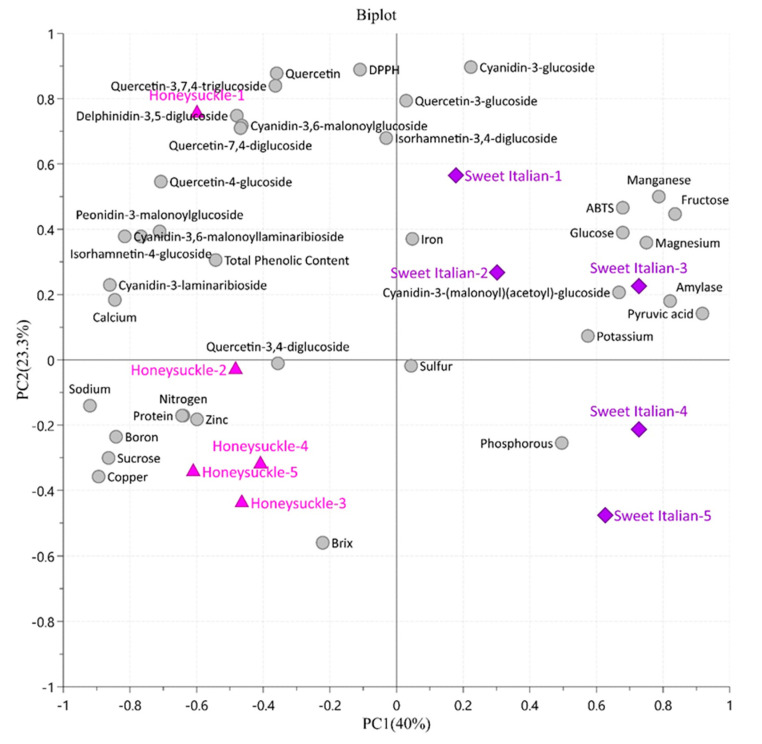
The biplot obtained from principal component analysis (PCA) describing the variation between honeysuckle and sweet Italian red onion cultivars and the correlation with the phytochemicals. The phytochemicals are represented as gray dots, and the cultivars are represented with the triangle (pink) and diamond (purple) markers.

**Table 1 plants-09-01077-t001:** The nutritional component of the red onions from two cultivars.

Nutritional Component	Honeysuckle	Sweet Italian
	μg/g	
Phosphorus	2525.63	2677.64
Potassium	12,720.67	13,550.1
Calcium	3183.54	2506.5
Magnesium	980.43	1100.62
Sodium	1001.34	314.13
Zinc	14.43	12.23
Iron	23.16	27.69
Copper	7.38	3.19
Manganese	6.03	9.22
Sulfur	3421.58	3415.67
Boron	18.34	15.34
	%	
Nitrogen	1.47	1.31
Protein	9.21	8.2

**Table 2 plants-09-01077-t002:** Identification of flavonoids and anthocyanins in red onions by LC-ESI-QTOF-MS using the positive-ionization mode.

Phytochemical	Compound	Experimental MS (*m*/*z*)	Theoretical MS (*m*/*z*)	Mass Error (ppm)	Aglycone Ion (*m*/*z*)
**Flavonoids**	Quercetin-3,7,4′-triglucoside	789.20136	789.208399	8.9	303.04816
	Quercetin-7,4′-diglucoside	627.15072	627.155576	7.7	303.04766
	Quercetin-3,4′-diglucoside	627.15279	627.155576	4.4	303.04919
	Isorhamnetin-3,4′-diglucoside	641.16577	641.171226	8.5	317.06311
	Quercetin-3- glucoside	465.10128	465.102753	3.1	303.04827
	Quercetin-4′-glucoside	465.10382	465.102753	−2.2	303.05104
	Isorhamnetin-4′-glucoside	479.12018	479.118403	−3.7	317.06622
	Quercetin	303.04965	303.049929	0.9	-
**Anthocyanins**	Cyanidin-3-glucoside	449.1022	449.1078	12.4	287.0516
	Cyanidin-3-laminaribioside	611.1549	611.1606	9.3	287.0524
	Delphinidin-3,5-diglucoside	627.1496	627.1555	9.4	303.0474
	Cyanidin-3-(6”-malonoylglucoside)	535.1024	535.1082	10.8	287.0531
	Cyanidin-3-(6”-malonoyl-laminaribioside)	697.1532	697.161	11.1	287.0527
	Peonidin-3-malonoylglucoside	549.1168	549.1238	12.7	301.0665
	Cyanidin-3(malonoyl)-(acetyl)-glucoside	577.1124	577.1187	10.9	287.0518

**Table 3 plants-09-01077-t003:** Quantification of flavonoids and anthocyanins in red onions by HPLC.

Phytochemical	Compound	Honeysuckle (μg/g FW)	Sweet Italian (μg/g FW)
**Flavonoids**	Quercetin-3,7,4′-triglucoside	8.44 ± 0.95	6.95 ± 0.47
	Quercetin-7,4′-diglucoside	35.34 ± 1.80	28.73 ± 1.16
	Quercetin-3,4′-diglucoside	168.91 ± 8.71	154.79 ± 3.91
	Isorhamnetin-3,4′-diglucoside	13.18 ± 1.56	13.42 ± 0.53
	Quercetin-3-glucoside	5.88 ± 0.34	6.47 ± 0.27
	Quercetin-4′-glucoside	184.60 ± 7.11 *	115.14 ± 3.35
	Isorhamnetin-4′-glucoside	25.58 ± 1.19 *	13.52 ± 0.64
	Quercetin	7.44 ± 0.59	6.31 ± 0.35
	Total Flavonoids	449.37 ± 75.21 *	345.34 ± 58.09
**Anthocyanins**	Cyanidin-3-glucoside	6.42 ± 0.94	9.30 ± 1.77
	Cyanidin-3-laminaribioside	5.34 ± 0.49 *	3.03 ± 0.67
	Delphinidin-3,5-diglucoside	1.25 ± 0.08	1.07 ± 0.19
	Cyanidin-3-(6”-malonoylglucoside)	58.88 ± 3.76	50.90 ± 6.79
	Cyanidin-3-(6”-malonoyl-laminaribioside)	30.18 ± 2.14 *	20.44 ± 2.84
	Peonidin-3-malonoylglucoside	1.01 ± 0.08 *	0.66 ± 0.12
	Cyanidin-3(malonoyl)-(acetyl)-glucoside	0.33 ± 0.09	1.05 ± 0.35*
	Total Anthocyanins	103.40 ± 22.06	86.47 ± 18.43

Values are mean ± standard deviation of three samples; * indicate significant difference between two cultivars as determined by students *t*-test at *p* < 0.05.

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
