# Peer review of "Comparative Metabolomics Profiling of Polyphenols, Nutrients and Antioxidant Activities of Two Red Onion (Allium cepa L.) Cultivars"

_plants, 2020, doi:10.3390/plants9091077_

Round 1

Reviewer 1 Report

Dear Authors, 

Thank you for the opportunity to read your work. The manuscript is well-written and interesting. But, I have some questions and comments for the Authors. 

1. Why you choose these cultivars of red onions? 

2. Are these cultivars most widely used only in the United States? What about worldwide use? 

3. Could you please add more information about these cultivars: main differences, advantages in agriculture, etc. 

4. Organically or conventionally these cultivars were grown? How do you think organically or conventionally growing may impact the composition and quantity of biologically active compounds (etc. phenolic compounds) in these cultivars?  

5. Line 195, please put a dot in the end. 

6. Line 341, please add more information about the freeze-drying process (time, equipment, etc.).

7. For protein and mineral analysis you used freeze-dried samples and for other analysis (antioxidant, total phenolics...) used fresh onions. Do you think the preparation (fresh, frozen, freeze-dried, oven-dried, etc.) of the onions could impact the composition and content of the predominant compounds and how? Could it be your future experiments? 

8. Line 406 please put a dot in the end. 

9. The authors identified the main flavonoids and anthocyanins in two cultivars of red onions: Quercetin-3,4′-diglucoside, quercetin-4′-glucoside, cyanidin-3-(6′′-malonoylglucoside). Are these compounds predominate in other cultivars of red onions? If not, what compounds are the main? Please add this information in the manuscript. 

10. Table 2, please correct the table text that words will be in one line (for example look at Phytochemical > Phytochemical (in one line whole word).  

Author Response

Please see the attached replies to reviewer 1

Reviewer 2 Report

Dear Editor,

I have carefully read the manuscript by Rita Metrani et al, submitted for publication in Plants.

In my opinion the paper is well written, the experiments are properly conducted and results clearly described.  Although the scientific relevance of research could be considered limited, the manuscript contains useful information for the readers and for this reason I think it merits publication in Plants.

I have only some minor points that need to be considered:

Line 33:  Ref 1:  In my opinion this reference does not report the pieces of information referred by the authors, instead the information is reported in the letterature cited therein.

Line 49:  anticholesterol

Line 64: They have anticancer ……

Line 65: They also improve cardiovascular health, and show

Line 67:  anthocyanins ………derivatives were also found

Line 70: which exert

Line 77: States

Line 133: oxidative reactions…………..performance of enzymes

Table 2 and 3: an editing of the tables to adapt column text is necessary

Table 3:  Quercetin     7.44 ± 0.59     6.31 ± 0.35  not 6.135

Lines 202 and 203: isorhamnetin

Line 209:  Serban et al

Line 235: molecules

Line 238: display

Line 242: two molecules of glucose

Line 243: the present peak was identified…..

Line 402 :  Please check if the content of  formic acid is  0.1 (or 1%?)

Author Response

Please see the attached replies to reviewer 2

Reviewer 3 Report

The article “Comparative metabolomics profiling of polyphenols,  nutrients and antioxidant activities of two red onion  (Allium cepa L.) cultivars” is interesting topic for Plants journal and after minor revision can be accepted for publishing.

Please, find some comments bellow.

Introduction part please add references for this line ”Health benefits of phenolics attract the researcher and breeders to identify and develop new cultivars with enhanced functional

properties”.

For example:

Sytar O. (2015) Phenolic acids in the inflorescences of different varieties of buckwheat and their antioxidant activity. Journal of King Saud University - Science, 27 (2), 136-142. Sytar, 2015 https://doi.org/10.1016/j.jksus.2014.07.001.

Kaushik, P., Andújar, I., Vilanova, S., Plazas, M., Gramazio, P., Herraiz, F. J., Brar, N. S., & Prohens, J. (2015). Breeding Vegetables with Increased Content in Bioactive Phenolic Acids. Molecules (Basel, Switzerland)20(10), 18464–18481. https://doi.org/10.3390/molecules201018464

In the phrase “Selection of the phenolic-rich cultivars, preferably the red ones with high antioxidant activity, can deliver a variety of health benefits [6]. “ Please, specify which plant in phrase “Selection of the phenolic-rich cultivars of onion....”. It will create connection with next paragraph.

L401 “Anthocyanins extraction of was performed according to published paper.” PLease, remove “of”.

In Conclusion or Discussion parts it would be good to add comparable information not just about studied onion cultivar but also with other red-coloured onion cultivars to show or say about nutritional potential of experimental cultivars.  

Author Response

Please see the attached replies to reviewer 3

Reviewer 4 Report

The authors present a study comparing the content of polyphenols and sugars, nutritional composition, pungency, antioxidant activity, and alpha-amylase inhibition ability between two red onion cultivars; the honeysuckle and sweet Italian. The methodology is strong and thorough, however, the presentation, connection to the overall literature and interpretation of results needs improvement. Below I present my recommendations to improve the work and make it more accessible to readers.

Introduction

  • The health benefits of polyphenols and their antioxidative action is described quite confusingly. For example, in the paragraph that starts on page 2, line 47, the authors should make clear which health benefits of flavonoids are the consequence of the antioxidative activity directly (radical scavenging); and which due to other mechanisms, such as binding to specific protein targets. See Molecules, 2016 Jul;21(7):901. The same should be done for anthocyanins.
  • Page 2, paragraph starting at line 69: Here you discuss the nutritional value of onions, where you also discuss their protein content. It would be beneficial if you could specify which amino acids are present in onions (essential versus non-essential). Moreover, I imagine the average consumption of onions is not significant for humans to obtain any noticeable levels of protein. Please discuss the onions intake in average human diets.
  • Page 2, paragraph starting at line 69: Please discuss which compounds specifically were already proven previously to have alpha-amylase inhibiting effects and what is their mechanism if known.

Results and Discussion

Pungency, total soluble solids and sugars

  • Please discuss the connection between pyruvic acid and pungency. Include previous literature, where this connection is described.

Evaluation of protein and minerals

  • It would be beneficial for readers if you could add the recommended daily intakes to Table 1 and also express the contents in weight. Moreover, the paragraph is confusing, as for some minerals you specify their physiological importance (e.g. potassium) while for others, you do not (e.g. phosphorous).

Evaluation of antioxidant activity, total phenolic content and alpha-amylase assay

  • Please discuss why you used DPPH as well as ABTS assay. How are they complementary?
  • Can you speculate which substances are responsible for the alpha-amylase inhibition?

Identification and quantification of flavonoids

  • Based on your study the main flavonoids in onions are quercetin and its glycosides. You only briefly discuss its potential health-promoting effects. I believe you should expand on this, and include the specific pharmacological activity of quercetin. See Nutrients, 2019 Feb;11(2):257. Do the same for anthocyanins (cyanidin).
  • Discuss the importance of glycosides in concern with their pharmacokinetics.
  • Ellagic acid has been found to offer various health benefits (Antioxidants. 2020 Jul;9(7):587.); and interestingly shows synergistic effects with quercetin, which onions are rich in (The Journal of nutrition. 2003 Aug 1;133(8):2669-74.). If onions contain enough ellagic acid (Annals of Agricultural Sciences. 2014 Jun 1;59(1):33-9.), this synergism could provide for an interesting discussion in the paper.

Principal component analysis and Pearson’s correlation coefficients

  • Page 13, line 302: Is there any existing literature reporting specifically on cyanidin(-3-glucoside) inhibiting alpha-amylase?

Methods and Materials

  • In your analysis, you used methanolic extracts. Please discuss in the paper, how this can influence your overall results, especially in concern to potential lipophilic compounds in onions.

Author Response

Please see the attached replies to reviewer 4

Round 2

Reviewer 4 Report

The authors successfully addressed all issues raised by this reviewer. Consequently, the manuscript has been significantly improved and can be in its current version recommended for publication in Plants.